# Effects of Repeated Injection of 1% Lidocaine vs. Radial Extracorporeal Shock Wave Therapy for Treating Myofascial Trigger Points: A Randomized Controlled Trial

**DOI:** 10.3390/medicina58040479

**Published:** 2022-03-26

**Authors:** Areerat Suputtitada, Carl P. C. Chen, Narin Ngamrungsiri, Christoph Schmitz

**Affiliations:** 1Department of Rehabilitation Medicine, Faculty of Medicine, Chulalongkorn University, and King Chulalongkorn Memorial Hospital, Bangkok 10330, Thailand; topnarin@outlook.com; 2Department of Physical Medicine and Rehabilitation, Chang Gung Memorial Hospital at Linkou, College of Medicine, Chang Gung University, Guishan District, Taoyuan City 33343, Taiwan; carlchendr@gmail.com; 3Chair of Anatomy II, Institute of Anatomy, Faculty of Medicine, LMU Munich, 80336 Munich, Germany; christoph_schmitz@med.uni-muenchen.de

**Keywords:** elasticity index, lidocaine, myofascial trigger points, pressure pain threshold, radial extracorporeal shock wave therapy, rESWT, upper trapezius muscle

## Abstract

*Background and Objectives*: This study tested the hypothesis that treatment of myofascial trigger points (MTrPs) in the upper trapezius muscle (UTM) with repeated injection of 1% lidocaine results in better alleviation of muscular stiffness and soreness as well as improved metabolism in the hypercontracted MTrP area than treatment with radial extracorporeal shock wave therapy (rESWT). *Materials and Methods*: A single-blinded, prospective, randomized controlled trial was conducted on patients suffering from MTrPs in the UTM. Thirty patients were treated with repeated injection of 2 mL of 1% lidocaine (three injections; one injection per week). Another 30 patients were treated with rESWT (three treatment sessions; one treatment session per week; 2000 radial extracorporeal shock waves per treatment session; positive energy flux density = 0.10 mJ/mm^2^). The primary outcome measure was pain severity using the VAS score. The secondary outcome measures included muscle elasticity index, pressure pain threshold and neck disability index. Evaluation was performed at baseline (T1), 15–30 min after the first treatment in order to register immediate treatment effects (T2), before the second treatment (i.e., one week after baseline) (T3) and one week after the third treatment (i.e., four weeks after baseline) (T4). *Results*: There were no statistically significant differences in the primary and secondary outcome measures between the patients in the lidocaine arm and the patients in the rESWT arm at T1 and T4. Within the arms, the mean differences of all outcomes were statistically significant (*p* < 0.001) when comparing the data obtained at T1 with the data obtained at T3 and the data obtained at T4. *Conclusions*: The results of this pilot study suggest that the use of rESWT in patients with MTrPs in the UTM is safe and leads to reduced pain and improved muscle elasticity, pressure pain threshold and neck disability index, without adverse effects. Larger trials are necessary to verify this. Clinicians should consider rESWT instead of injections of lidocaine in the treatment of MTrPs in the UTM.

## 1. Introduction

Myofascial pain syndrome (MPS) is a condition characterized by local and referred pain as well as autonomic symptoms, which are all produced by myofascial trigger points (MTrPs). The most widely accepted hypothesis regarding the pathogenesis of MTrPs is sustained sarcomere contraction due to excessive acetylcholine release at the neuromuscular junction on the basis of overuse or muscle injury [1,2,3,4,5,6]. Pain is produced by the contracted muscle tissue compressing the blood vessels, causing local ischemia and vasoneuroactive release, especially at the site of the MTrPs [1,2,3,4,5,6]. The neurogenic inflammatory effect and tissue swelling lead to an energy crisis, obstruction of calcium intake to the sarcoplasmic reticulum and prevention of sarcomere shrinkage. Some studies utilized microdialysis (i.e., intramuscular biochemical analysis without taking biopsies) to understand the biological characteristics of MTrPs [7,8], and found that the pH in MTrPs was lower than in the surrounding muscle tissue. Furthermore, compared to latent MTrPs or normal muscle tissue, active MTrPs were demonstrated to contain significantly higher levels of bradykinin, substance *p*, tumor necrosis factor, interleukin (IL)-1, IL-6, IL-8, calcitonin gene-related peptide (CGRP), serotonin and norepinephrine [7,8,9].

Upon palpation, active MTrPs induce spontaneous discomfort, referred pain and motor or autonomic symptoms, including diminished range of motion, muscular weakness and lack of coordination [1,2,3,4,5,6,8]. Latent MTrPs present these characteristics to a much lesser degree when palpated and/or compressed [1,2]. On the other hand, systematic reviews and meta-analyses of studies on manual palpation of MTrPs discovered small sample sizes and variations in inter- and intra-rater reliability [10,11].

Imaging techniques such as ultrasonography (US), magnetic resonance elastography (MRE) and ultrasono-elastography (UE) were used to study the physical characteristics of taut bands of skeletal muscle fibers and MTrPs [12,13,14]. These studies reported increased taut band rigidity in MRE images, significantly reduced vibration amplitude of affected tissue by UE, changes in blood vessel systolic and diastolic velocities near MTrPs by color doppler US, and focal hypoechoic areas with heterogeneous echotexture of MTrPs when imaged with two-dimensional US (2D US) [12,13,14]. Due to higher tissue stiffness, UE offers the ability to quantify the viscoelastic properties of the tissue, allowing objective identification of MTrPs. Elasticity variations can be detected as color variance in elastograms. Some equipment can calculate the strain ratio (SR) between two regions of interest (ROI), allowing imaging findings to be quantified and reference values to be provided. Only a few studies used UE to assess MPS and MTrPs so far, but demonstrated more objective diagnosis of MTrPs and improved monitoring of treatment success compared with other diagnostic measures [12,13,14].

Repeated injection of local anesthetics is an established procedure for treating MTrPs [15,16,17]. However, these treatments are invasive and potentially myotoxic [18,19]. During the last decades, extracorporeal shock wave therapy (ESWT) has emerged as an effective and safe alternative for treating MTrPs [20,21,22,23,24,25,26,27,28,29,30,31,32,33,34,35,36,37]. According to a recent systematic review and meta-analysis, ESWT appears to be associated with higher pain alleviation than sham ESWT or ultrasound therapy in treatment of MTrPs in the UTM [38]. Most importantly, there was no statistically significant difference in pain intensity or neck disability index when compared the outcome of ESWT to the outcome of conventional therapies (dry needling, trigger point injection, laser therapy) [38].

This study tested the hypothesis that treatment of MTrPs in the UTM with repeated injection of 1% lidocaine results in better alleviation of muscular stiffness and soreness as well as improved metabolism in the hypercontracted MTrP area than treatment with radial ESWT (rESWT). Treatment success was assessed using objective (elasticity index (EI)) and subjective (visual analogue score (VAS) of pain, pressure pain threshold (PPT)) outcome measures.

## 2. Materials and Methods

This study was a randomized, controlled trial (RCT) with blinded assessors and statisticians. The study was carried out in compliance with the Declaration of Helsinki of the World Medical Association (WMA). It was approved by the Institutional Review Board of the Faculty of Medicine, Chulalongkorn University, Bangkok, Thailand (Protocol No.: 222/57; date of approval: 17 September 2015), and was registered with the Thai Clinical Trials Registry (Identifier TCTR20160330003). At any time, all patients had the option to withdraw their informed consent to participate in this study. There was no commercial support for this research.

Patients were enrolled in this study from January 2016 through June 2019 at the Department of Rehabilitation Medicine, Faculty of Medicine, Chulalongkorn University, and King Chulalongkorn Memorial Hospital, Bangkok, Thailand. Patients with MTrPs who were diagnosed by A.S. using the criteria specified in [6] were considered eligible for this study. The inclusion criteria were as follows: (1) age between 20 and 25 years; (2) diagnosis of only one active MTrP in the UTM on either the left or the right side; (3) mild to moderate pain intensity at baseline (VAS pain between 3 and 6; with VAS = 0 representing no pain at all and VAS = 10 representing maximum, intolerable pain); (4) ability to attend the hospital during the treatment and follow-up assessments; and (5) willingness to sign the informed consent form. The exclusion criteria were as follows: (1) fixed contractures or deformities of the shoulder and neck; (2) diseases of bones and joints; (3) clinical signs of myopathy and neuropathy; (4) treatment of MTrPs in the UTM with injection of lidocaine, ESWT, injection of any other local anesthetics or Botulinum neurotoxin, dry needling, drugs or any other treatment during a period of three months before inclusion in this study; (5) previous surgery of the shoulder and neck; (6) epilepsy; (7) intellectual disability; (8) infection, tumor, ulcer, or skin condition at the treatment site.

Figure 1 depicts the flow of patients through this study according to the Consolidated Standards of Reporting Trials (CONSORT) [39]. Eligible patients were randomly assigned to either treatment with injection of lidocaine or treatment with rESWT by means of a computer-generated random numbers list. No patient was lost to follow-up, allowing for a complete analysis of all patients at all follow-up examinations.

Neither the UE outcome assessor nor the outcome assessor for pain severity (VAS, PPT) and the neck disability index (NDI) were blinded to group randomization, and did not participate in the implementation of interventions. The statistician was also uninformed about the group assignment.

Injections of lidocaine and rESWT were administered by a single, skilled physiatrist (A.S.).

Patients in the lidocaine arm received intramuscular injections of 2 mL of 1% lidocaine at the site of the MTrP using a 25 Gauge needle (three injections, one injection per week). A local twitch reaction when the trigger site was injected confirmed the MTrP.

Patients in the rESWT arm were treated with a Swiss DolorClast rESWT device (Electro Medical Systems, Nyon, Switzerland), EvoBlue handpiece and 15-mm convex applicator (three treatment sessions; one treatment session per week; 2000 radial extracorporeal shock waves (rESWs) per treatment session; air pressure of the device set at 2.5 bar, resulting in a positive energy flux density of 0.1 mJ/mm^2^; frequency of the rESWs set at 12 Hz).

In addition, all patients were taught a simple home exercise routine of UTM stretching during the first visit. The patients received a video file of the UTM stretching exercise on their mobile phones. Patients had to finish 10 sessions of UTM stretching twice a day. For ethical concerns, patients were allowed to take acetaminophen or nonsteroidal anti-inflammatory drugs (NSAIDs) if the discomfort became intolerable, depending on the pain severity. The amount of acetaminophen and NSIADs was recorded.

The primary outcome measure was pain severity using the VAS score. The secondary outcome measures included muscle elasticity index (EI), pressure pain threshold (PPT) and neck disability index (NDI). Evaluation was performed at baseline (T1), 15–30 min after the first treatment session in order to register immediate treatment effects (T2), before the second treatment session (i.e., one week after baseline) (T3) and one week after the third treatment session (i.e., four weeks after baseline) (T4).

The VAS score was assessed by letting the patients select a point on a 10 cm scale ranging from 0 (no pain at all) to 10 (maximum, intolerable pain) that best reflected their level of pain at the time of assessment. A clinically significant difference in pain severity was represented by a mean reduction in VAS of 2.0 cm, which correlates to effective therapy [40,41,42].

The UE elasticity index was acquired using a LOGIQ S7 Expert ultrasound scanner (GE Healthcare, Little Chalfont, UK), with ML6-15 high-frequency linear ultrasound transducer, and Elasto-Q dedicated quantitative software, Power, intensity and edge enhancement were set before data collection for image correction, and were maintained the same for all pre- and post-treatment assessments (a lower EI implies better elasticity of the muscle). After the patient’s head was placed in a neutral position, the UTM was detected using the 2D ultrasound mode of the device. Then the examiner performed an elasticity index measurement of the UTM by rhythmic tissue compression and decompression, adhering to the equipment criteria for standardization of the process (c.f. [12,13,14]). Finally, elastograms were generated using the Elasto-Q software, which simultaneously captured and displayed 2D ultrasound and elastogram images of the same region of interest (ROI). After identifying the active MTrP in the UTM, three ROIs representing the active MTrP were selected for localizing the best sinusoidal compression. Then, the mean EI was determined as shown in Figure 2. All UE data were collected by a blinded expert ultra-sonographer (N.N.), and were independently analyzed by another blinded expert ultra-sonographer (C.C.).

A digital pressure algometer (Model PTH AF2; Pain Diagnostics and Thermography, Great Neck, NY, USA) was used to determine the least amount of pressure that was required to cause pain [43]. To this end, the rubber tip area of the algometer was perpendicularly pressed on the MTrPs at a rate of 1 kg/s. When the patient started to feel pain, the compression was quickly stopped (i.e., the higher the PPT, the better the patient can tolerate pressure on the muscle). Three repeated measurements were performed at each spot at intervals of 30 s, and the average value was determined.

The NDI is a widely used self-rating scale for assessing how pain affects everyday activities [44]. Pain severity, personal care, lifting objects, reading, headache, focus, work, driving, sleeping, and amusement are among the 10 sections of the NDI. Participants were asked to assign a score to each section that best represented the average during the previous week. A minimal clinically significant difference of 3.5 points was proposed in the literature [44] (the lower the NDI, the better the patient feels).

Any unexpected events that happened throughout the experiment were recognized and recorded as adverse effects.

Statistical analysis of all data was performed using the Statistical Program for Social Sciences (SPSS) version 17 (SPSS Inc., Chicago, IL, USA), with *p* values smaller than 0.05 considered statistically significant. Independent t-tests or Mann–Whitney U tests for continuous variables (depending on normality) were used to compare baseline characteristics between the arms; Fisher’s exact test was used to compare baseline characteristics for the categorical variable Gender between the arms. A 2 × 3 repeated measures analysis of variance was used on the outcome measures, with time (T1, T2, T3, and T4) as the within-subject component and treatment (lidocaine, rESWT) as the between-subject factor. When the sphericity assumption was broken, the Greenhouse-Geisser correction was used. The significance values were adjusted using the Bonferroni correction.

The study’s power was calculated based on the assumption that 70% of the patients in the Lidocaine arm would achieve a reduction in VAS pain by more than 2 cm at T4, but only 30% of the patients in the rESWT arm (note that these calculations were performed in 2015, i.e., at a time when studies on treatment of MTrPs of the UTM with the rESWT device used in this study were not published). With a type I error rate of 5%, a power of 80%, and at least 10% follow-up losses, 25 patients per arm were required. We expanded the sample size to 30 patients per arm to strengthen the power of this study. The power analysis was performed with the tool, OpenEpi [45].

## 3. Results

### 3.1. Characteristics of the Patients at Baseline

The characteristics of the patients at baseline (T1) are summarized in Table 1. There were no statistically significant differences between the arms at baseline.

### 3.2. Outcomes

Table 2 shows all outcome measures and the results of the comparisons between the arms; Table 3 summarizes the results of the within-arms comparisons between T1 and T3 as well as between T1 and T4. To make the results easier to understand, they are also graphically presented in Figure 3.

Compared to rESWT, injection of 1% lidocaine resulted in statistically significantly smaller mean VAS score, mean EI, mean PPT and mean NDI at T2 (i.e., immediately after the first treatment) and T3 (i.e., before the second treatment), but not at T4 (i.e., four weeks after baseline). Within the arms, both treatments resulted in statistically significantly smaller mean VAS score, mean EI, mean PPT, and mean NDI at T3 and T4 compared to T1.

### 3.3. Adverse Effects

Pain and reddening of the skin were noted in a few cases, but the patients did not drop out. There were no severe adverse effects and no long-term negative consequences.

### 3.4. Need for Additional Treatments

No patient experienced pain that would have required acetaminophen or NSAIDs.

## 4. Discussion

A recent systematic review and meta-analysis on treatment of myofascial pain syndrome with ESWT [38] listed eight studies addressing neck and upper back pain [21,26,29,30,32,34,35,36]. All of these studies demonstrated efficacy and safety of ESWT in treatment of MPS of the neck and upper back, with reductions in mean VAS scores that were generally comparable with the outcome of this study. However, except for one study [21] injection of trigger points was not applied in these studies. Unfortunately, in the latter study [21] it was not described what was actually injected, which renders the results of [21] difficult to interpret. Furthermore, injection of trigger points was combined with transcutaneous electrical nerve stimulation (TENS) (three weeks, five times per week, duration of 20 min each) in [21]. The authors found no difference between the treatments.

Another recent study compared rESWT with rESWT + injection of 3 mL of 0.5% lidocaine (three treatment sessions, one treatment session per week) [37]. The authors found that rESWT alone statistically significantly reduced the mean VAS pain from baseline to four weeks post baseline, and the combination of rESWT with injections of lidocaine resulted in a slightly better outcome than rESWT alone that was also statistically significant [41]. These data may indicate that the combination of rESWT and injections of lidocaine into MTrPs may be superior to rESWT alone in treatment of MTrPs of the UTM. On the other hand, one has to consider that no data have been published demonstrating the energy flux density achieved with the rESWT device used in [37] (Powershocker LGT-2500S Plus; International Electro Medical Company, Dehli, India [46]). All that is known about this device is that it can be operated at an air pressure between 1 bar and 5 bar and a frequency of the rESWs between 1 and 22 Hz [46]. However, this information is not useful considering that different handpiece technologies of rESWT devices can result in very different energy flux densities when operated at the same air pressure, and most probably all rESWT devices except of the one used in our study substantially lose energy with increasing frequency of the rESWs applied [47]. The reason is a fundamental difference in the handpiece technology used in the EvoBlue handpiece of the Swiss DolorClast device, which is protected by a patent [48]. This technology keeps the energy flux density of the rESWs generated using the EvoBlue handpiece of the Swiss DolorClast almost the same over the entire frequency range (1–20 Hz in the device used in this study). Accordingly, the results reported in [37] must not be interpreted such that the combination of rESWT and injections of lidocaine into MTrPs may always result in better outcome than rESWT alone in treatment of MTrPs of the UTM. Rather, the results reported in [37] could demonstrate that rESWT performed with insufficient energy flux density may require additional injections of lidocaine in order to be come as effective as rESWT performed with sufficient energy flux density in treatment of MTrPs of the UTM.

Treatment of MTrPs with ESWT was also applied in a number of other studies that were not considered in the systematic review and meta-analysis performed in [20,22,23,24,25,27,28,31,33,37,38]. However, except for one study [24] ESWT was not compared to injections of lidocaine (or other local anesthetics) in these studies. In [24] patients received two injections of 0.2 mL of 0.3% lidocaine per MTrP per week for a total of four weeks (i.e., a total of eight injections). This translates into 3.3% of the amount of lidocaine that was applied in this study. Unfortunately, this study [24] has two major issues: (i) the authors did not report when their single follow-up examination was performed; and (ii) the statistical analysis performed (one way ANOVA; post-hoc tests not described) is inadequate for a study that compared the effects of three different treatments (injections of lidocaine, ESWT and proprioceptive neuromuscular facilitation) before and after the treatments. Accordingly, the conclusions drawn in [24] should be handled carefully. In any case, the percent reduction in mean VAS pain reported in [24] (lidocaine arm: −56.3%; rESWT arm: −54.3%) was much less than what was found in this study (lidocaine arm: −82.4%; rESWT arm: −81.6%).

Our study had two major results: (1) All outcome measures were statistically significantly improved after the first injection of lidocaine compared to the situation after the first rESWT session (Table 2 and Figure 3). Accordingly, if one is primarily interested in initial pain relief as fast as possible in treatment of MTrPs of the UTM, injection of lidocaine may be the first choice. However, this comes at the price of an invasive therapy with the risk of bacterial and viral contamination of the treated muscle, as well as potential myotoxicity [18,19]. (2) Four weeks after baseline we did not find any statistically significant difference in the outcome measures between rESWT and injections of lidocaine in treatment of MTrPs of the UTM (Table 2 and Figure 3). As rESWT does not share the disadvantages of injections of lidocaine (rESWT is not invasive, does not cause potential bacterial and viral contamination of the treated muscle, and is not myotoxic), rESWT appears to be the better choice if one is interested in effective and safe treatment of MTrPs of the UTM with lasting effects. The latter is indicated by results of a recent study on treatment of functional and structural muscle injuries in professional athletes using the same rESWT device that was used in this study [49].

Altered flexibility of the UTM following rESWT could be one of the mechanisms mediating the beneficial effects of rESWT on MTrPs. However, this is only one of many potential mechanisms that may explain the outcome of this study. Other potential mechanisms are briefly summarized in the following. (i) rESWT improved the NDI, which is consistent with previous findings [21,27,36]. One may speculate that this was caused at least in part by mechanical separation of actin and myosin filaments in MTrPs. (ii) Exposure of muscle tissue to rESWs may result in temporary disruption of nerve transmission at the neuromuscular junction [50,51]. (iii) Substance *p*, a pain neurotransmitter, may be abundant in active MTrPs [7], and exposure of tissue to extracorporeal shock waves was demonstrated resulting in lower substance *p* levels [52]. (iv) According to the gate control theory of pain [53], mechanical impact of rESWs on large-diameter afferent fibers may alter the transmission of nerve impulses from small-diameter afferent fibers to spinal cord transmission cells in the dorsal horn. (v) Reduced lubricin expression is involved in the pathophysiology of MTrPs, and ESWT may help by increasing lubricin expression in septa [54]. (vi) Another mechanism for pain relief could be increased muscle microcirculation as a result of repetitive ESWT [55]. Deciphering the exact molecular and cellular mechanisms resulting in reduced VAS pain and improved EI, PPT and NDI would require to take muscle biopsies and/or performing microdialysis, which may be performed in future studies. Such studies may also include a subsequent trial of combined rESWT and lidocaine injection, and employing rESWT for 2000 rESWs per treatment session as performed in this study, to account for both needle effect, lidocaine effect, and rESWT regeneration effects in the same session.

Collectively, the studies on ESWT for MTrPs of the UTM published so far [20,21,22,23,24,25,26,27,28,29,30,31,32,33,34,35,36,37] did not show any advantage of focused ESWT (fESWT) over rESWT or vice versa (a detailed discussion of differences between and similarities of fESWT and rESWT is provided in [56]). This is important because the use of fESWT is restricted to physicians in many countries (as is the case in Germany where chiropractors and physiotherapists who have been trained in Germany are not legally entitled to use fESWT). Moreover, the International Society for Medical Shockwave Treatment (ISMST) has recommended that only a qualified physician (certified by National or International Societies) may use fESWT in their latest Consensus Statement regarding ESWT indications and contraindications [57]. However, in many countries a medical doctor may not be available to perform treatment of MTrPs of the UTM as performed in this study. This is also the reason why e.g., a randomized controlled trial on acute Type 3b hamstring muscle injuries currently being undertaken is based on rESWT rather than on fESWT [58].

Our study has some limitations. First, the patients were not blinded to the therapy, which may have caused some subjective bias. Second, there were no subgroups with variable rESWT intensities, intervals, or frequencies because total energy is dose-dependent, which could alter therapy benefits and cost-effectiveness. Identifying appropriate regimens and conducting cost-effective assessments should be a priority in the future. Third, this study was not a non-inferiority trial. Accordingly, this study could not establish that rESWT is as effective as injections of lidocaine in treatment of MTrPs of the UTM. Fourth, this study’s target demographic was individuals aged 20 to 25 years, avoiding MPS in combination with degenerative joint issues, which is typical in the elderly.

## 5. Conclusions

The results of this pilot study suggest that the use of rESWT in patients with MTrPs in the UTM is safe and leads to reduced pain and improved muscle elasticity, pressure pain threshold, and neck disability index, without adverse effects. Larger trials are necessary to verify this. Clinicians should consider rESWT instead of injections of lidocaine in the treatment of MTrPs in the UTM.

## Figures and Tables

**Figure 1 medicina-58-00479-f001:**
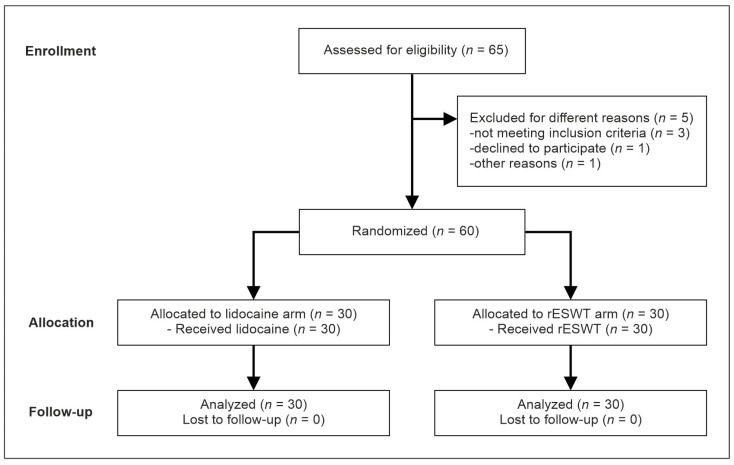
Flow of patients through this study according to CONSORT [39].

**Figure 2 medicina-58-00479-f002:**
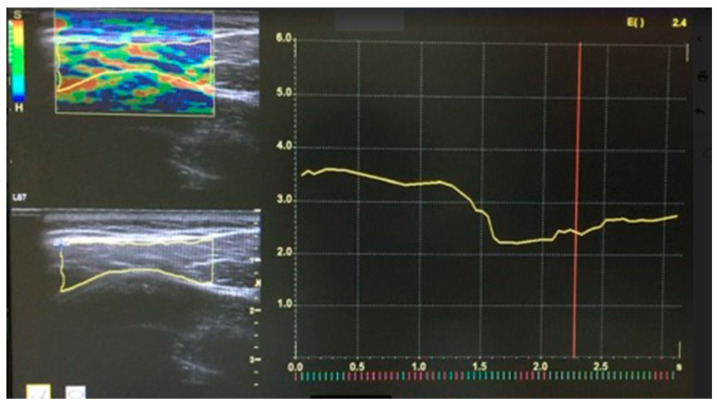
Sonoelastrogram of the upper trapezius muscle.

**Figure 3 medicina-58-00479-f003:**
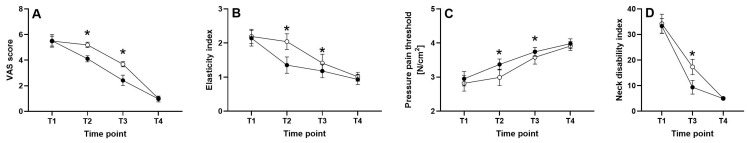
(**A**) Mean and standard deviation of the VAS score of the patients in the Lidocaine arm (closed dots) and the patients in the rESWT arm (open dots). (**B**) Mean and standard deviation of the elasticity index of the patients in the Lidocaine arm (closed dots) and the patients in the rESWT arm (open dots). (**C**) Mean and standard deviation of the pressure pain threshold of the patients in the Lidocaine arm (closed dots) and the patients in the rESWT arm (open dots). (**D**) Mean and standard deviation of the neck disability index of the patients in the Lidocaine arm (closed dots) and the patients in the rESWT arm (open dots). In all panels the asterisks indicate statistically significant differences (*p* < 0.05) between the arms (c.f. Table 2). Abbreviations: T1, baseline; T2, immediately after the first treatment; T3, before the second treatment (i.e., one week after baseline); T4, one week after the third treatment (i.e., four weeks after baseline).

**Table 1 medicina-58-00479-t001:** Characteristics of the patients at baseline (mean ± standard deviation).

Characteristics	Lidocaine Arm (*n* = 30)	rESWT Arm (*n* = 30)	*p*
Age [Y]	22.93 ± 1.03	23.05 ± 1.02	0.846
Gender [M:F]	12:18	13:17	>0.999
BMI [kg/m^2^]	23.36 ± 1.82	23.69 ± 1.37	0.857
VAS [cm]	5.51 ± 0.49	5.49 ± 0.39	0.621
EI	2.14 ± 0.23	2.19 ± 0.21	0.571
PPT [N/cm^2^]	2.95 ± 0.21	2.82 ± 0.23	0.728
NDI	33.3 ± 2.93	34.2 ± 3.69	0.619

Abbreviations: Y, years; M, male; F, female; BMI, body mass index; VAS, visual analogue scale; EI, elasticity index; PPT, pressure pain threshold; NDI, neck disability index.

**Table 2 medicina-58-00479-t002:** Outcome measures at T1, T2, T3, and T4, and comparison between arms (post hoc analysis).

Parameter (Time)	Lidocaine Arm		rESWT Arm		Comparison between Arms		
	Mean ± SD	95% CI	Mean ± SD	95 CI	D	*p*	95% CI
VAS (T1)	5.51 ± 0.49	4.82–6.12	5.49 ± 0.39	5.12–5.99	0.192	0.621	0.18–0.25
VAS (T2)	4.11 ± 0.24	3.78–4.36	5.19 ± 0.21	4.79–5.44	1.314	0.041 *	1.21–1.52
VAS (T3)	2.42 ± 0.41	2.02–2.84	3.68 ± 0.20	3.57–3.79	1.262	0.035 *	1.10–1.42
VAS (T4)	0.97 ± 0.23	0.76–1.11	1.01 ± 0.13	0.96–1.15	0.124	0.729	0.01–0.16
EI (T1)	2.14 ± 0.23	1.84–2.21	2.19 ± 0.21	2.08–2.23	0.148	0.571	0.10–0.17
EI (T2)	1.35 ± 0.24	1.67–1.99	2.04 ± 0.23	1.92–2.17	0.628	0.003 *	0.47–0.83
EI (T3)	1.18 ± 0.20	0.74–1.34	1.41 ± 0.26	1.26–1.55	0.371	0.034 *	0.29–0.43
EI (T4)	0.93 ± 0.15	0.78–1.21	1.01 ± 0.13	0.96–1.15	0.131	0.856	0.09–1.14
PPT (T1)	2.95 ± 0.21	2.70–3.23	2.82 ± 0.23	2.57–3.24	0.113	0.728	0.09–0.16
PPT (T2)	3.37 ± 0.16	3.11–3.64	2.99 ± 0.24	2.67–3.35	0.203	0.047 *	0.19–0.23
PPT (T3)	3.74 ± 0.13	3.52–3.93	3.58 ± 0.20	3.44–3.71	0.293	0.041 *	0.22–0.30
PPT (T4)	3.98 ± 0.14	3.71–4.16	3.91 ± 0.13	3.67–4.15	0.112	0.075	0.10–0.15
NDI (T1)	33.3 ± 2.9	31.1–35.1	34.2 ± 3.7	30.8–37.4	1.247	0.619	1.20–1.29
NDI (T2)	No data	No data	No data	No data	No data	No data	No data
NDI (T3)	9.34 ± 2.67	7.84–12.2	17.3 ± 2.99	14.7–19.8	6.912	0.041 *	3.13–9.15
NDI (T4)	4.91 ± 0.25	4.13–5.46	5.01 ± 0.50	4.96–5.45	0.151	0.891	0.15–0.19

Abbreviations: CI, confidence interval; D, difference of mean data; EI, elasticity index; NDI, neck disability index; PPT, pressure pain threshold; rESWT, radial extracorporeal shock wave therapy; T1, baseline; T2, immediately after the first treatment; T3, before the second treatment (i.e., one week after baseline); T4, one week after the third treatment (i.e., four weeks after baseline); VAS, visual analog scale; * significantly different statistically.

**Table 3 medicina-58-00479-t003:** Comparison of outcome measures within the arms (post hoc analysis).

Parameter	Arm	T1 vs. T3			T1 vs. T4		
		D	*p*	95% CI	Diff of Mean	*p*	95% CI
VAS	Lidocaine	2.21 ± 0.24	0.018 *	2.01–2.45	3.91 ±0.35	0.025 *	3.54–4.31
	rESWT	1.78 ± 0.23	0.014 *	1.85–2.32	4.01 ± 0.23	0.027 *	3.87–4.27
EI	Lidocaine	0.97 ± 0.18	0.024 *	0.67–1.21	1.17 ± 0.12	0.035 *	1.01–1.35
	rESWT	0.64 ± 0.15	0.031 *	0.59–0.84	1.13 ± 0.41	0.041 *	0.98–1.52
PPT	Lidocaine	0.84 ± 0.13	0.031 *	0.63–1.24	1.02 ± 0.13	0.043 *	0.84–1.17
	rESWT	0.69 ± 0.25	0.025 *	0.42–0.92	1.19 ±0.22	0.039 *	0.97–1.41
NDI	Lidocaine	22.2 ± 2.47	0.035 *	19.5–25.7	5.56 ± 0.94	0.038 *	4.62–6.11
	rESWT	28.4 ± 3.12	0.029 *	25.1–31.7	28.9 ± 3.18	0.032 *	24.4–32.2

Abbreviations: CI, confidence interval; D, difference of mean data; EI, elasticity index; NDI, neck disability index; PPT, pressure pain threshold; rESWT, radial extracorporeal shock wave therapy; T1, baseline; T3, before the second treatment (i.e., one week after baseline); T4, one week after the third treatment (i.e., four weeks after baseline); VAS, visual analog scale; * significantly different statistically.

## Data Availability

The data presented in this study are available on request from the corresponding authors.

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
