# Peer review of "Effects of Repeated Injection of 1% Lidocaine vs. Radial Extracorporeal Shock Wave Therapy for Treating Myofascial Trigger Points: A Randomized Controlled Trial"

_medicina, 2022, doi:10.3390/medicina58040479_

Round 1
Reviewer 1 Report
I admire the authors for exploring into this complex topic. MPS is a common disorder, and it's important to study and improve therapy options for it. There do not appear to have been many research comparing 1 percent lidocaine trigger point injections to rESWT and value added by objectively measured with elastic index. I decide to accept this novelty with minor revision.
A recent study published this year (2022) found that treating with rESWT + lidocaine injection (3 sessions, once a week) was superior to treating with rESWT (3 sessions, once a week) after four weeks. However, the authors used only 1000 shot of rESWT each session. (Anwar N, Li S, Long L, Zhou L, Fan M, Zhou Y, Wang S, Yu L. Combined effectiveness of extracorporeal radial shockwave therapy and ultrasound-guided trigger point injection of lidocaine in upper trapezius myofascial pain syndrome. Am J Transl Res. 2022 Jan 15;14(1):182-196.)
The suggestion in 4.1 should include a subsequent trial of combined rESWT and lidocaine injection, and employing rESWT for 2000 shots as this study to account for both needle effect, lidocaine effect , and rESWT regeneration effects in the same session.
Author Response
16 March 2022
Dear Editor and Reviewer 1,
We thank you and the reviewers for the comments and suggestions. Please find our point-by-point responses below.
We hope that our responses are satisfactory. Thank you very much for your kind consideration.
Best Regards,
Professor Dr. Areerat Suputtitada, MD.
On behalf of the authors
Point-by-point reply to the comments by Reviewer 1
I admire the authors for exploring into this complex topic. MPS is a common disorder, and it's important to study and improve therapy options for it. There do not appear to have been many research comparing 1 percent lidocaine trigger point injections to rESWT and value added by objectively measured with elastic index. I decide to accept this novelty with minor revision.
We are grateful for this comment by the reviewer.
A recent study published this year (2022) found that treating with rESWT + lidocaine injection (3 sessions, once a week) was superior to treating with rESWT (3 sessions, once a week) after four weeks. However, the authors used only 1000 shot of rESWT each session. (Anwar N, Li S, Long L, Zhou L, Fan M, Zhou Y, Wang S, Yu L. Combined effectiveness of extracorporeal radial shockwave therapy and ultrasound-guided trigger point injection of lidocaine in upper trapezius myofascial pain syndrome. Am J Transl Res. 2022 Jan 15;14(1):182-196.)
We have incorporated this study in the Discussion section of our revised manuscript as follows (Lines 269-294):
"Another recent study compared rESWT with rESWT + injection of 3 ml of 0.5% li-docaine (three treatment sessions, one treatment session per week) [41]. The authors found that rESWT alone statistically significantly reduced the mean VAS pain from baseline to four weeks post baseline, and the combination of rESWT with injections of lidocain resulted in a slightly better outcome than rESWT alone that was also statistically significant [41]. These data may indicate that the combination of rESWT and in-jections of lidocaine into MTrPs may be superior to rESWT alone in treatment of MTrPs of the UTM. On the other hand, one has to consider that no data have been published demonstrating the energy flux density achieved with the rESWT device used in [41] (Powershocker LGT-2500S Plus; International Electro Medical Company, Dehli, India). All what is known about this device is that it can be operated at an air pressure be-tween 1 bar and 5 bar and a frequency of the rESWs between 1 and 22 Hz [42]. How-ever, this information is not useful considering that different handpiece tech-nologies of rESWT devices can result in very different energy flux densities when op-erated at the same air pressure, and most probably all rESWT devices except of the one used in our study substantially lose energy with increasing frequency of the rESWs applied [43]. The reason is a fundamental difference in the handpiece technology used in the EvoBlue handpiece of the Swiss DolorClast device, which is protected by a patent [44]. This technology keeps the energy flux density of the rESWs generated using the EvoBlue handpiece of the Swiss DolorClast almost the same over the entire frequency range (1-20 Hz in the device used in this study). Accordingly, the results reported in [41] must not be interpreted such that the combination of rESWT and injections of li-docaine into MTrPs may always result in better outcome than rESWT alone in treat-ment of MTrPs of the UTM. Rather, the results reported in [41] could demonstrate that rESWT performed with insufficient energy flux density may require additional injec-tions of lidocaine in order to be come as effective as rESWT performed with sufficient energy flux density in treatment of MTrPs of the UTM."
The suggestion in 4.1 should include a subsequent trial of combined rESWT and lidocaine injection, and employing rESWT for 2000 shots as this study to account for both needle effect, lidocaine effect , and rESWT regeneration effects in the same session.
We have considered this suggestion by the reviewer in the Discussion section of our revised manuscript as follows (Lines 343-346):
"Such studies may also include a subsequent trial of combined rESWT and lidocaine in-jection, and employing rESWT for 2000 rESWs per treatment session as performed in this study, to account for both needle effect, lidocaine effect, and rESWT regeneration effects in the same session."
Reviewer 2 Report
Overall Summary:
The manuscript entitled “Effects of Radial Extracorporeal Shockwave Therapy Versus 1% Lidocaine 2 Injection for Myofascial Trigger Points Measured with Elastic Index”, presents the results of a randomized controlled trial to compare the effects of radial extracorporeal shockwave therapy (rESWT) to lidocaine injection for alleviation of myofascial trigger points in the trapezius.
Global Comments
The research topic is original and adds to the knowledge of the field by presenting novel findings regarding a treatment for trapezius MPS. This article is very well written and adds to our current understanding of … I have listed specific comments for the authors relative to each section of the manuscript.
Methods
- Please indicate who diagnosed the MPS.
Results
- Were there any differences in baseline characteristics (i.e., age, gender, BMI) across the groups? If not, please state or include p values in Table 1 to indicate results of comparisons.
- Line 212, says that there were no statistically significant differences between groups (…P=0.045…). It appears that PPT was different at TP4. See Table 2 as well.
- It would be helpful to interpret the outcomes (differences) in the Results section. For example, greater values on VAS indicate more pain, yet greater values for PPT indicate less pain sensitivity. Therefore, the interpretation of the Diff of Mean for VAS pain and PPT are opposite?? And what about EI? Are higher values better/worse? Same for NDI.
Discussion
- Line 248, again P=0.045 is described as non-significant. Please revise.
Author Response
16 March 2022
Dear Editor and Reviewer 2,
We thank you and the reviewers for the comments and suggestions. Please find our point-by-point responses below.
We hope that our responses are satisfactory. Thank you very much for your kind consideration.
Best Regards,
Professor Dr. Areerat Suputtitada, MD.
On behalf of the authors
Point-by-point reply to the comments by Reviewer 2
The manuscript entitled “Effects of Radial Extracorporeal Shockwave Therapy Versus 1% Lidocaine 2 Injection for Myofascial Trigger Points Measured with Elastic Index”, presents the results of a randomized controlled trial to compare the effects of radial extracorporeal shockwave therapy (rESWT) to lidocaine injection for alleviation of myofascial trigger points in the trapezius.
Global Comments
The research topic is original and adds to the knowledge of the field by presenting novel findings regarding a treatment for trapezius MPS. This article is very well written and adds to our current understanding of
We are grateful for this comment by the reviewer.
… I have listed specific comments for the authors relative to each section of the manuscript.
Methods
Please indicate who diagnosed the MPS.
All patients were diagnosed by the first author of this study, Dr. Areerat Suputtitada.
We have added this information in the Methods section of our revised manuscript (Line 104).
Results
Were there any differences in baseline characteristics (i.e., age, gender, BMI) across the groups? If not, please state or include p values in Table 1 to indicate results of comparisons.
There were no differences in baseline characteristics between the patients in the Lidocaine arm and the patients in the rESWT arm.
We did not perform statistical analyses of baseline characteristics in our initial manuscript because of the following statement in the updated CONSORT guidelines for reporting parallel group randomised trials (Moher et al., Int. J. Surg. 2012;10:28-55):
"Unfortunately significance tests of baseline differences are still common; they were reported in half of 50 RCTs trials published in leading general journals in 1997. Such significance tests assess the probability that observed baseline differences could have occurred by chance; however, we already know that any differences are caused by chance. Tests of baseline differences are not necessarily wrong, just illogical. Such hypothesis testing is superfluous and can mislead investigators and their readers. Rather, comparisons at baseline should be based on consideration of the prognostic strength of the variables measured and the size of any chance imbalances that have occurred."
Anyway, because the reviewer requested tests of baseline differences, we have added the corresponding P values in Table 1 in our revised manuscript.
Line 212, says that there were no statistically significant differences between groups (…P=0.045…). It appears that PPT was different at TP4. See Table 2 as well.
We are grateful for this comment by the reviewer. We went back to our raw data and found that the 95% confidence interval of the PPT obtained for the rESWT arm and the P value of the difference of the mean PPT values at T4 were not correct in Tabe 2 of our initial manuscript.
We have corrected these typos in our revised manuscript. In addition, we carefully checked all data provided in the tables and the main text of our revised manuscript, but did not find additional typos in this regard.
It would be helpful to interpret the outcomes (differences) in the Results section. For example, greater values on VAS indicate more pain, yet greater values for PPT indicate less pain sensitivity. Therefore, the interpretation of the Diff of Mean for VAS pain and PPT are opposite?? And what about EI? Are higher values better/worse? Same for NDI.
We have provided the corresponding explanations in our revised manuscript (Lines 153, 161-162, 182-183 and 190).
Discussion
Line 248, again P=0.045 is described as non-significant. Please revise.
Please see above. This typo was corrected in our revised manuscript (Table 2)
Because this reviewer judged both the description of the methods and the presentation of the results as "can be improved" in the online Review Report Form, we edited all sections of the manuscript for achieving a better representation of our study, obviously without changing the content or the results of our study. We also transferred our manuscript into the style of the Word template of the journal, Medicina.